# Development of a Novel, Easy-to-Prepare, and Potentially Valuable Peptide Coupling Technology Utilizing Amide Acid as a Linker

**DOI:** 10.3390/ph17080981

**Published:** 2024-07-24

**Authors:** Yaling Wang, Fan Yang, Hongyan Li

**Affiliations:** 1College of Chemistry and Materials Science, Fujian Normal University, Fuzhou 350117, China; xmwangyaling@fjirsm.ac.cn; 2Xiamen Institute of Rare Earth Materials, Haixi Institute, Chinese Academy of Sciences, Xiamen 361021, China; 3Fujian Institute of Research on the Structure of Matter, Chinese Academy of Sciences, Fuzhou 350002, China; 4Fujian Science & Technology Innovation Laboratory for Optoelectronic Information of China, Fuzhou 350108, China; 5Xiamen Key-Laboratory of Rare Earth Photoelectric Functional Materials, Xiamen 361021, China; 6Key Laboratory of Rare Earths, Chinese Academy of Sciences, China Rare Earth Group Research Institute, Ganzhou 341000, China; 7Fujian Province Joint Innovation Key Laboratory of Fuel and Materials in Clean Nuclear Energy System, Fujian Institute of Research on the Structure of Matter, Chinese Academy of Sciences, Fuzhou 350002, China; 8Key Laboratory of Rare Earths, Ganjiang Innovation Academy, Chinese Academy of Sciences, Ganzhou 341000, China; 9Xiamen Key Laboratory of Rare Earth Photoelectric Functional Materials, Xiamen Institute of Rare Earth Materials, Haixi Institute, Chinese Academy of Sciences, Xiamen 361021, China; 10Department of Medical Physics, Institute of Modern Physics, Chinese Academy of Sciences, Lanzhou 730000, China

**Keywords:** radionuclide coupling drugs, peptides, bifunctional chelating agents, nuclear magnetic resonance (NMR), chromatography–mass spectrometry (LC-MS)

## Abstract

The process of synthesizing radionuclide-coupled drugs, especially shutdown technology that links bipotent chelators with biomolecules, utilizes traditional coupling reactions, including emerging click chemistry; these reactions involve different drawbacks, such as complex and cumbersome reaction steps, long reaction times, and the use of catalysts at various pH values, which can negatively impact the effects of the chelating agent. To address the above problems in this study, This research designed a novel bipotent chelator coupled with peptides. In the present study, dichloromethane was used as a solvent, and the reaction was conducted at room temperature for 12 h. A one-step ring-opening method was employed to introduce the coupling functional group of tridentate amide acid. The coupling materials consisted of the amino active site of the peptide and diethylene glycol anhydride. In this paper, this study explored the reactions between different equivalents of acid anhydride coupled to the peptide (peptide sequence: HLRKLRKR) and determined that the maximum conversion of the peptide feedstock was 87%. To determine the selectivity of the reaction sites in this polypeptide, This study identified the peptide sequence at the reaction site using nuclear magnetic resonance (NMR) and liquid chromatography–mass spectrometry (LC-MS). For the selected peptide, the first reactive site was on the terminal amino group, followed by the amino group on the tetra- and hepta-lysine side chains. The tridentate amic acid framework functions as a chelating agent, capable of binding a range of lanthanide ions. This significantly reduces and optimizes the time and cost associated with synthesizing radionuclide-coupled drugs.

## 1. Introduction

Peptide–drug conjugates (PDCs) have become among the most well-researched and commercially successful types of drugs at home and abroad [1]. In the last 50 years, especially in the last 20 years, the research and development of radionuclide drug conjugates (RDCs), another similar type of coupled drug design, has surged due to the rapid development and wider application of nuclear medicine and nuclear technology [2]. A special type of coupled drugs, RDCs are used to deliver different radionuclides to tumor cells for diagnostic and therapeutic purposes, and these conjugates utilize molecular carriers specific to tumor antigens [3]. Radionuclide-coupled drugs are structurally composed of carrier molecules that mediate targeting, linker arms, bifunctional chelating agents (BFCAs), and radioisotopes [4,5]. The carrier molecules generally consist of biologically active molecules such as peptides, antibodies, etc.

In the structure of RDCs, a pharmacokinetic modification linker (PKM) is usually used to bridge BFCAs and the carrier molecule; this linker helps to modify the pharmacokinetic profile of the whole complex and creates space between the carrier and the radionuclide chelator complex [6]. The bifunctional chelator is covalently linked to the carrier molecule. This connection must be stable under physiological conditions and must not significantly impair the binding strength and specificity of the carrier molecule [7]. Considering the predominance of primary amines and free thiols on many biomolecules, the following bond types constitute the vast majority of chelator–biomolecule linkages: amide, thiourea, and thioether bonds [8]. For clinical reference, most of the production of radiolabeled peptides relies primarily on following traditional coupling chemistries: (1) the formation of peptide bonds through reactions of primary amines and carboxylic acids activated with succinimidyl esters (NHS), sulfosuccinimidyl esters (SNHS), tetrafluorophenol (TFP), or peptide-coupling reagents (e.g., HATU, HOBT, etc.); (2) the modification of cysteine by a maleimide-based reagent (thiosulfate) to form thioether bonds [9]; and (3) the functionalization of lysine to prepare NH_2_ side chains by using N-hydroxysuccinimide (-NHS) or isothiocyanate (-NCS) reagents [10,11,12]. Conventional coupling chemistry involves several problems, as biomolecules often have a variety of free primary amines and thiols, leading to the nonspecific and selective linkage of bifunctional chelators to biomolecules, in addition, the reaction steps are complex and cumbersome; the reaction time is too long; or the reaction pH is detrimental to the biologically active molecules and affects the stabilization of the complexes in the organism. However, these are not the only options for conjugation reactions, and the emergence of “click chemistry” has provided a new biorthogonal approach for attaching bifunctional chelators to biomolecules [13]. The main characteristics of click chemistry are the presence of high yields, absence or harmlessness of byproducts, high selectivity, and the need for a high thermodynamic drive [14]. Although click chemistry was originally applied in organic chemistry, this reaction can be used in radiopharmaceutical chemistry for the introduction of radionuclides. In particular, short-lived radionuclides such as ^11^C and ^18^F are introduced. Click chemistry encompasses a range of highly efficient reactions, such as Michael addition reactions [15] involving thiols, ring-opening reactions of nitrogen- and oxygen-containing cyclic ions (e.g., azetidinium ions and epoxides), and the conversion reactions of aldehydes to hydrazones and oximes. These reactions are fast, highly selective, and essential for the synthesis of complex molecules [16]. Huisgen cycloaddition (CuAAC) is the most classical system for click chemistry reactions [17], whereby azide-to-terminal or -internal alkynes undergo a monovalent copper-catalyzed 1,3-dipole cycloaddition, which occurs between azides and the terminal or internal alkyne; this reaction is catalyzed by monovalent copper to generate 1,2,3-triazoles, and the resulting triazole heterocycle mimics an amide bond, which can assist in metal chelation [18]. However, the method has several limitations because click chemistry is based on stringent criteria. For example, one of the greatest disadvantages of basic CuAAC is that copper is used as a catalyst, which is associated with potential toxicity. Although the human body requires copper for certain functions, an excess of copper can have serious consequences [19]. Click chemistry also significantly impacts drug discovery and drug delivery for the synthesis of materials with desirable kinetic properties.

With so many methods of coupling available, it becomes important to choose one that is appropriate for the task at hand. Various chelating agents necessitate distinct coupling methods. Many different chelating agents have been used for zirconium to date, and desferrioxamine B (DFO) has been the most successful and commonly used chelating agent [20]. In most cases, the N-suc-DFO or DFO-Bz-NCS methods are sufficient. Octadentate macrocyclic chelators, such as DOTA (1,4,7,10-tetra-azacyclododecane-1,4,7,10-tetraacetic acid), have been developed for labeling ^90^Y, ^177^Lu, and radioactive lanthanides, exhibiting superior stability properties compared to acyclic ligands. The most common approach is to attach the target peptide to one of the four acetate groups of DOTA via a CO-NH bond. This coupling reaction can be carried out using active esters of carboxylic acids, such as NHS esters [21]. DTPA (diethylenetriaminepentaacetic acid) is an excellent acyclic chelator that provides stable and rapid chelation kinetics. It has been successfully utilized for radioactive metals such as ^64^Cu, ^111^In, ^177^Lu, and ^86/90^Y [22,23,24]. Coupling peptides can be achieved by using dihydrogenated DTPA or tri-t-butyl-DTPA as a chelator.

In summary, both traditional coupling chemical reactions and emerging click chemistry have various drawbacks, including harsh reaction conditions, complex reaction steps, and the use of harmful catalysts. When linking various chelating agents to biologically active molecules like peptides and antibodies, it is essential to select the suitable coupling reaction. This undoubtedly adds complexity to the synthesis of radionuclide-conjugated drugs. In order to address these issues, we developed a novel coupling technique with peptides using amide acid as a linker in this study. We utilized the amino active site of peptides or antibodies (mono-antibody, bi-antibody) and diethylene glycol anhydride as the coupling material to introduce the coupling functional group of tridentate amide acid [25]. The reaction is characterized by the easy availability of raw materials, simple conditions, selectivity for the peptide’s active site, and high reaction yield, and it does not require the participation of metal catalysts, thus avoiding cytotoxicity and the impact on the bifunctional chelating agent. In addition, when diethylene glycol anhydride successfully couples with the peptide, a compound with a diethylene glycol–amino acid backbone is formed. This compound, with a tridentate amide acid backbone, was initially used as a “green” extractant for lanthanides. This unique backbone structure provides the compounds with an excellent chelating ability for lanthanide ions, surpassing other types of carboxylic acid extractants [25,26,27]. Therefore, when the peptide is coupled with diethylene glycol anhydride, the tridentate amide backbone can act as a chelating agent for a variety of lanthanide ions. This significantly reduces both the time and cost required for synthesizing radionuclide-coupled drugs.

## 2. Results and Discussion

### 2.1. Establishment of Peptide Standard Curves

Eight peptide standard solutions (2.5, 2, 1.5, 1, 0.75, 0.5, 0.25, 0.125 mg/mL) were analyzed according to the elution gradient in the peptide standard curve, and the standard curve was plotted. The linear relationships between the peak area and the concentration and between the slope and intercept of the curve were determined using the standard curve. The results showed that the linear relationship of the standard curve was good when the mass concentration of the peptide was within 2.5–0.125 mg/mL, and the linear equation of the standard curve was y = 16,494.39x − 822.59, and had a correlation coefficient of R^2^ > 0.99. The peptides showed a good linear relationship within the concentration range of the assay, i.e., there was a significant positive correlation between the peak area and the concentration [28].

### 2.2. Degree by Which Different Equivalents of Anhydride Are Coupled to Peptides

We selected different equivalents of anhydride for the coupling reaction with the peptide, and the equivalence ratios of the anhydride to the peptide were 10:1, 20:1, 40:1, and 60:1, in which the anhydride was in excess. The reacted system was post-treated and subjected to liquid chromatography to generate a chromatogram, as shown in Figure 1.

As shown in Figure 1, the number of peaks is identical for the different equivalence ratios of the liquid chromatograms even though the equivalents of anhydride were different; the area of each peak changes as the anhydride equivalence increases. Each peak was collected and identified by LC-MS. The first peak has a molecular weight of 1105, which is the raw material peak of the peptide. The second peak has a molecular weight of 1221 and is the product peak of a peptide molecule coupled to an anhydride. The molecular weights of the third peak and the fourth peak are 1337, which are generated by a peptide molecule coupled to two anhydride molecules, and only the reaction sites are different. The fifth peak has a molecular weight of 1453 and is generated by a peptide molecule coupled to three anhydride molecules.

Table 1, Table 2, Table 3 and Table 4 show the changes in the peak time and peak area of each peak in the reactions between the peptide and anhydride with different equivalents. The first peak corresponds to the raw polypeptide, and the area of the raw polypeptide peak decreases with increasing anhydride concentration. When the anhydride equivalent was increased to 40 and 60, the areas of the raw peptide peaks varied slightly. The peak area was taken into the standard curve of the peptide, and the conversion rate of the raw material for each equivalence ratio could be obtained, as shown in Table 5. When the ratio of acid anhydride to peptide reached 60:1, the conversion rate of the raw material no longer increased significantly, and reached equilibrium, with a maximum conversion rate of 87%.

As the amount of anhydride increased, the peak area of the feedstock peptide decreased until an equilibrium was reached. The peak corresponding to the anhydride of the peptide product first increased and then decreased when 40 equivalents of anhydride were added, but the peak area of the peptide product was the highest at this time, indicating that the highest yield of the substance was generated. The peak was observed to increase and then decrease, except for the peptide product coupled to an anhydride. This mainly occurs because the other reaction sites on the product become coupled to more anhydrides when the amount of anhydride continues to increase, and anhydrides are transformed into other coupled products. The peak areas of all other products increased with increasing anhydride equivalents.

### 2.3. Determination of the Peptide Reaction Site

The sequence of the reactive active sites was progressively determined, and the products were subjected to NMR spectroscopy. For ease of illustration and analysis, the relevant carbon and hydrogen atoms of the histidine in the raw peptide are labeled here, as shown in Figure 2 below. Figure 3, Figure 4, Figure 5, Figure 6 and Figure 7, respectively, show the ^1^H NMR spectra of raw peptides, ^13^C NMR spectrum, ^1^H-^13^C HSQC NMR spectrum, ^1^H-^1^H COSY NMR spectrum, and ^1^H-^13^C HMBC NMR spectrum. 

In the chemical analysis of the imidazole ring, the hydrogen spectrum (Figure 3) and the ^1^H-^13^C HSQC spectrum (Figure 4) provided this paper with the key information to understand its structural composition in a detailed and precise manner. This paper was able to identify the precise data of the proton signals, i.e., δH values of 8.61 (1H, s, H-2) and 7.27 (1H, s, H-4), respectively. Similarly, the carbon signals were also clearly recorded at 134.8 ppm (C-2) and 117.6 ppm (C-4), respectively. These signals reveal the presence of carbon–carbon and hydrogen–hydrogen bonds in the imidazole molecule.

In the ^1^H-^1^H COSY spectra (Figure 5), a clear correlation signal between H-7 (4.17 ppm) and H-6 (3.23/3.12 ppm) can be observed.

In the detailed analysis of the ^1^H-^13^C HMBC spectra (Figure 6), the researchers observed a series of distinct and consistent chemical signals. Among these signals, the significant correlation between the H-4 region and the C-2 and C-5 (128.7 ppm) molecules suggests that there may be a specific structural or functional link between them. Further, the H-7 region also exhibited a clear correlation with the neighboring C-5/C-8 (167.7 ppm) region, whereas the H-6 region showed a stable correlation with C-7 (51.8 ppm) as well as the C-8/C-5/C-4 fragment. These multiple correlations confirm the existence of the aforementioned fragments.

The chemical shift value in the carbonyl or stacked alkene region is typically defined as 150 ppm, but it commonly exceeds 165 ppm. When the chemical shift value surpasses 200 ppm, it usually indicates the presence of aldehydes and ketones. In the analyzed carbon spectral data, eight specific carbonyl carbon signals can be clearly identified in this paper, which are 173.4, 172.3, 172.2, 171.6, 171.4, 171.3, 171.2, and 167.7. These signals indicate precise matches to the corresponding structural units in the raw material.

Based on the information provided, the significant chemical shifts in this polypeptide chain are identified in this paper, as shown in Figure 8.

After analyzing the NMR spectra of the raw peptide, this paper proceeds to examine the product following the reaction, which is a peptide coupling product linked to a diethylene glycol anhydride product. It was originally analyzed by LC-MS (liquid chromatography–mass spectrometry) that the terminal amino group of the peptide was coupled to a diethylene glycol anhydride molecule. Because this reaction site is the first to initiate the coupling of the peptide, it plays a crucial role in the selectivity of the coupling reaction. In this paper, the structure of the product was further verified through NMR hydrogen spectroscopy.

For ease of analysis, this paper labels the relevant positional behavior of the carbon atoms at the coupling site, as illustrated in Figure 9. The labeling of the histidine-associated carbon–hydrogen atoms in the coupling product is as described above and shown in Figure 2.

The hydrogen spectrum (Figure 10), along with the ^1^H-^13^C HSQC spectrum (Figure 11), can determine that the diethylene glycol anhydride has reacted with -NH_2_. The ^1^H-^13^C HMBC spectra show significant correlation signals between H-3 and C-2/C-4, and H-2 and C-1/C-3. The -NH_2_ of the peptide is involved in the reaction, forming the amide active hydrogen proton -NH with a chemical shift of 8.13 ppm. The proton signals in the imidazole ring were δH 8.68 (1H, s, H-2) and 7.22 (1H, s, H-4), while the carbon signals were at 134.1 ppm (C-2) and 116.8 ppm (C-4), respectively.

In the HMBC spectrum (Figure 12), the active hydrogen proton is observed to have a clear correlation signal with C-1, and the peak splits into a doublet peak, suggesting a linkage to the -CH hypomethyl group in the peptide. In addition, H-4 shows a significant correlation signal with C-2/C-5 (130.2 ppm) [29].

In the ^1^H-^1^H COSY spectra (Figure 13), a clear correlation signal between H-7 (4.62 ppm) and H-9 (8.13 ppm) can be observed.

In the detailed analysis of carbon spectra (Figure 14), a series of distinctive signals for carbonyl carbon were observed in this paper. These signals appear at 10 specific positions: 173.4, 172.4, 172.1, 171.6, 171.5, 171.3, 171.2, 171.1, 170.0, and 169.2. Through an in-depth study and comparison of these signals, it can be confirmed that they match the chemical structures present in the products.

Comparison with the raw material revealed that the amino group at position 9 in the product disappeared and an amide was formed, which appeared as an amide–active hydrogen proton signal at 8.13 ppm and cleaved into a d-peak; these results suggested that the group was linked to the -CH hypromethyl group.

Compared with that of the raw material, the proton signal of the 7-position hypomethyl group changed from 4.17 ppm to 4.62 ppm (a change of 0.45 ppm), suggesting that -NH_2_ may have reacted by connecting the electron-withdrawing group, resulting in enhanced deshielding and a large chemical shift.

In the HHCOSY spectrum obtained for the product, the -NH at position 9 is significantly correlated with H-7, indicating that -NH_2_ is involved in the reaction.

In addition to performing NMR spectroscopic analyses of the peptide raw material and the peptide products coupled to an anhydride molecule, we determined the peptide sequence of all coupled peptide products, and the following results were obtained.

The chemical shift values of important structures in peptide products coupled with an anhydride molecule are shown in Figure 15.

In addition to performing NMR spectroscopic analyses of the peptide raw material and the peptide products coupled to an anhydride molecule, we determined the peptide sequence of all coupled peptide products; the specific results are shown in Table 6. Table 6 shows that the first reaction site of the whole peptide is the terminal amino group of the peptide. After the terminal amino group is coupled to an anhydride, the other sites continue to react with the anhydride via a ring-opening reaction. The next site of reaction is the amino group on the lysine side chain of the peptide at the tetra- and hepta-positions, which are almost identical. As the anhydride equivalent increases, all three reaction sites on the peptide couple to the anhydride molecule.

## 3. Materials and Methods

### 3.1. Materials

The selected peptide sequence was HLRKLRKR, which was customized by Kingsley Bioscience & Technology. Unless otherwise stated, all other reagents were purchased from commercial sources, and the companies purchasing the reagents were mainly Aladdin Reagent (Shanghai, China) Co. and Sinopharm Chemical Reagent Co. (Shanghai, China). The NMR spectra of hydrogen spectrum, carbon spectrum, hhcosy, HMBC, and HSQC information were obtained on an AVANCE III HD 500 MHz (Karlsruhe, Germany) instrument and analyzed using MestNova software 6.1.0. HPLC was performed on an Agilent 1260 Infinity instrument with UV spectroscopy at 214 nm (Agilent Technologies, Wilmington, DE, USA), and the Lablogic Flow-Count detector was a Bioscan Co. B-FC-3200 photomultiplier detector (Bioscan Inc., Washington, DC, USA), and was analyzed using Laura 1.6 software (LabLogic Systems Ltd., Sheffield, UK). The brand and specifications of the column were a Shimadzu reversed-phase C-18 column (Shim-pack Scepter C18-120, 5 μm, 4.6 × 250 mm, PIN:227-31020-06, S/N:116FB20376). The mobile phase used for the analytical reversed-phase HPLC consisted of phase A with 0.1% TFA in water and phase B with 0.1% TFA in acetonitrile. LC/MS data were obtained on a Zorbax 300SB-C18 peptide trap (Agilent Technologies, Wilmington, DE, USA) series liquid chromatograph, and the mobile phases for LC/MS were A: water and 0.1% formic acid and B: a mixture of MeCN and 0.1% formic acid.

### 3.2. Establishment of High-Performance Liquid Chromatographic Standard Curves for Peptides

The peptide with the custom sequence HLRKLRKR was accurately weighed, and 25 mg of peptide standard was dissolved and concentrated to 10 mL using pure aqueous solution to generate a concentration of 2.5 mg/mL for the standard solution. The standard was diluted step by step with pure water to the required mass concentrations of 2, 1.5, 1, 0.75, 0.5, 0.25, and 0.125 mg/mL and injected into the liquid chromatograph. The chromatogram was recorded, and linear regression was carried out with the peak area as the vertical coordinate and the concentration as the horizontal coordinate. The standard curve of high-performance liquid chromatography can be plotted directly in the origin.

The chromatographic conditions were as follows: Shimadzu reversed-phase C-18 column (Shim-pack Scepter C18-120, 5 µm, 4.6 × 250 mm, PIN:227-31020-06, S/N:116FB20376), column temperature 30 °C, and 0.1% trifluoroacetic acid solution as mobile phase A and 0.1% trifluoroacetic acid in acetonitrile as mobile phase B. The separation was carried out on a Shimadzu reversed-phase C-18 column with the following gradient:0–25 min, with a linear gradient from 10% to 35% in liquid B;25–30 min with a linear gradient from 35% to 10% in liquid B.

### 3.3. Coupling Reactions of Diglycolic Anhydride with Peptides

The peptide sequence we selected was HLRKLRKR, which consists of four amino acids, L-histidine, leucine, L-lysine, and L-arginine, and the presence of multiple amino groups on the side chain of this peptide also results in the presence of multiple reaction sites. The optimal feeding ratio between the reactants of this coupling reaction, as well as the selectivity of the reaction sites, were explored. The structural structure of the polypeptide with associated equations is shown in Figure 16.

We designed different equivalents of diethylene glycol anhydride for the coupling reaction with the peptide. We selected peptide to anhydride equivalence ratios of 1:10, 1:20, 1:40, and 1:60, with the anhydride being in excess. The specific steps were as follows:The peptide (2 mg, 1.8 × 10^−3^ mmol) (peptide sequence HLRKLRKR) was dispensed in EP tubes with 0.5 mL of dichloromethane.Then, different equivalents of diethylene glycol anhydride were added to the solution in step 1, and the reaction was stirred for 12 h at room temperature.Postreaction treatment: The reacted system was frozen with liquid nitrogen and placed in a freeze-dryer to remove excess solvent.

Next, 0.5 mL of purified water was added to the reaction system in which excess solvent was removed, and the prepared solution was filtered through a 0.22 μm membrane and injected into a high-performance liquid chromatograph to monitor the extent of the reaction. The chromatographic conditions and elution gradient were the same as those established for the peptide standard curve.

### 3.4. Peptide Sequencing

Sample preparation: Samples were purified by HPLC, desalted by ultrafiltration (10 KD), and detected by mass spectrometry.

Chromatographic separation: Liquid A used in the liquid phase was a 0.1% formic acid aqueous solution, and liquid B was a 0.1% formic acid acetonitrile aqueous solution (84% acetonitrile). The liquid chromatography column (0.15 mm × 150 mm, RP-C18, Column Technology Inc., Fremont, CA, USA) was equilibrated with 95% liquid A. The samples were uploaded by an autosampler to Zorbax 300SB-C18 peptide traps (Agilent Technologies, Wilmington, DE, USA) and then separated on a column with the relevant liquid phase gradient set as follows:0–50 min, with a linear gradient from 4% to 50% in liquid B;50–54 min with a linear gradient from 50% to 100% in liquid B;for 54–60 min, fluid B was maintained at 100%.

Mass spectrometry identification: The ultrafiltration desalination products were separated by capillary high-performance liquid chromatography (HPLC) and then analyzed by mass spectrometry on a Q Exactive HF-X mass spectrometer (Thermo Fisher, Waltham, MA, USA). The analysis time was 60 min, and the detection mode was positive ions. The mass–charge ratios of the peptides and peptide fragments were determined according to the following method: 10 fragment profiles were collected after each full scan (MS2 scan).

Data analysis: Raw files of mass spectrometry tests (from the raw files) were searched in the corresponding databases using the software MaxQuant 1.5.5.1.

### 3.5. NMR Experimental Methods

The solvent used for NMR analysis is DMSO-D6 ((CD_3_)_2_S=O). A range of 20–25 mg of the sample was dissolved in this solvent and analyzed on a Bruker 500 MHz NMR spectrometer using five different modes: ^1^H experiment, ^13^C experiment with decoupling, ^1^H-^1^H COSY experiment, ^1^H-^13^C multiplicity edited HSQC with gradient selection BF1 <= 600 MHz, and ^1^H-^13^C HMBC with gradient selection, and these five modes were scanned and tested.

## 4. Conclusions and Outlook

Here, this work designed a novel technique for coupling bifunctional chelators to peptides. This study used the amino active site of the peptide or antibody (mono-antibody or bi-antibody) and diethylene glycol anhydride as the coupling material, and introduced the coupling functional group of the tridentate amide acid. In this study, various equivalents of anhydrides were chosen for the coupling reaction with peptides. When the ratio of anhydrides to peptides reached 60:1, the conversion of raw materials was 87%. This work used correlation spectra from NMR and determined peptide sequences to identify the sequence of reactions at these relevant sites for the selected polypeptides. The first reactive site of the entire peptide is the terminal amino group of the peptide. After the terminal amino group is coupled to an anhydride, the other sites continue to react with the anhydride via a ring-opening reaction. The next site of reaction is the amino group on the lysine side chain of the peptide at the tetra- and hepta-positions, which are almost identical. As the anhydride equivalent increases, all three reaction sites on the peptide couple to the anhydride molecules. The introduced tridentate amide acid skeleton, serving as a chelator with multiple coordination sites, can form stable complexes with lanthanide elements, certain nuclear energy metals, and the heavy metal lead (Pb) [25,30,31,32,33,34,35].

In future research, a comprehensive series of experiments will be conducted on synthesized peptide couplers to thoroughly investigate their properties and potential. Initially, researchers will conduct a chelation reaction using radioactive lanthanide elements like ^177^Lu to achieve a high radiochemical yield. Subsequently, in vitro cytotoxicity tests and animal studies will evaluate the biocompatibility and safety of these radioactively labeled peptide couplers. These rigorous experiments underscore the significance of radioactive elements and offer crucial data to support clinical applications in realistic biological settings.

## Figures and Tables

**Figure 1 pharmaceuticals-17-00981-f001:**
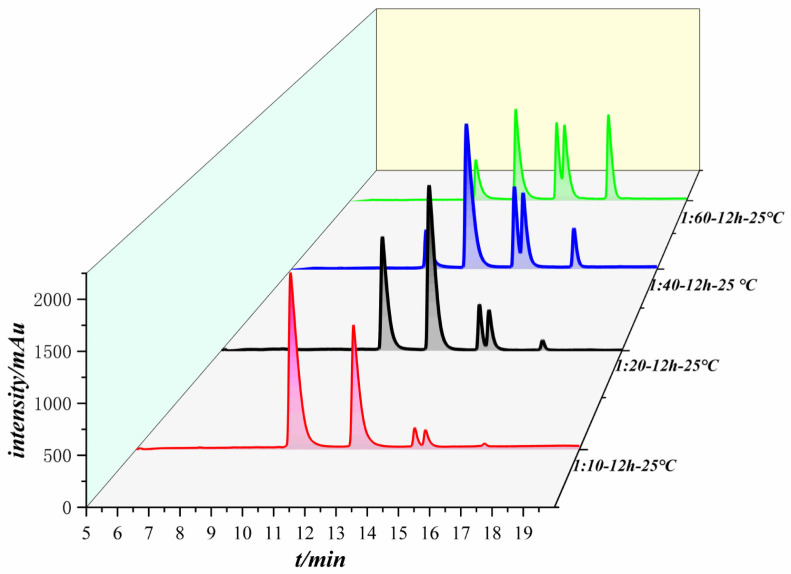
Liquid chromatograms of the reactions of different equivalents of acid anhydride with peptides.

**Figure 2 pharmaceuticals-17-00981-f002:**
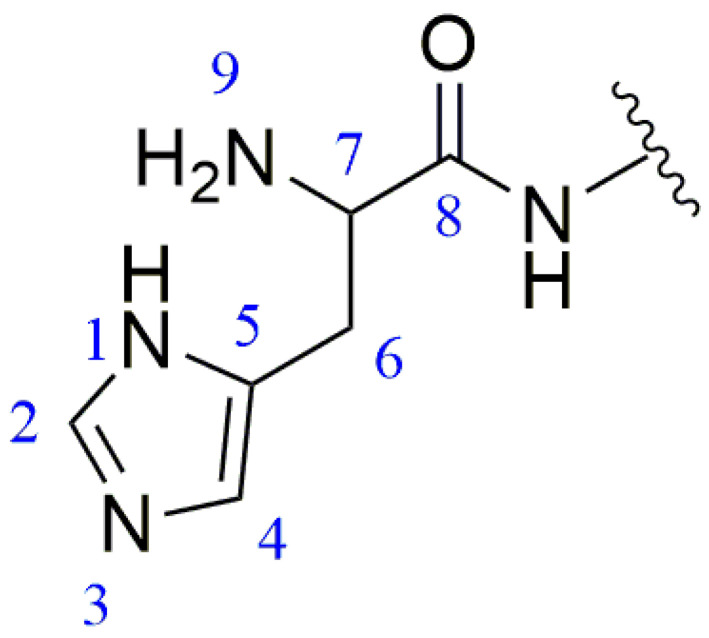
Histidine site number.

**Figure 3 pharmaceuticals-17-00981-f003:**
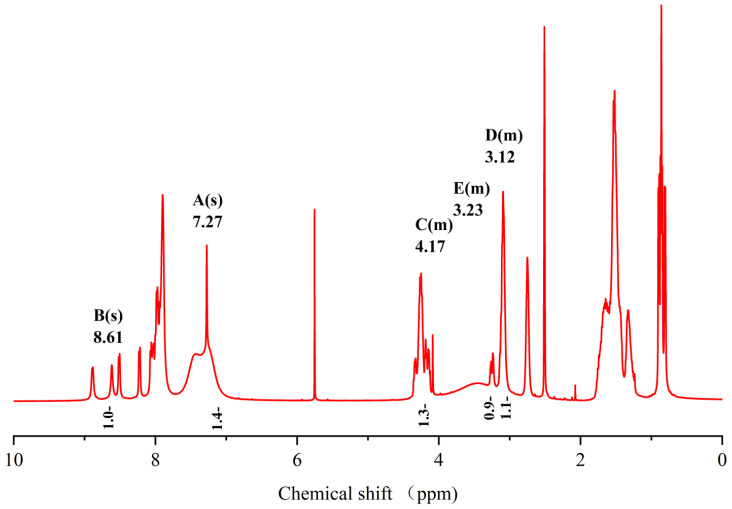
Spectrum of ^1^H NMR spectrum of raw peptides.

**Figure 4 pharmaceuticals-17-00981-f004:**
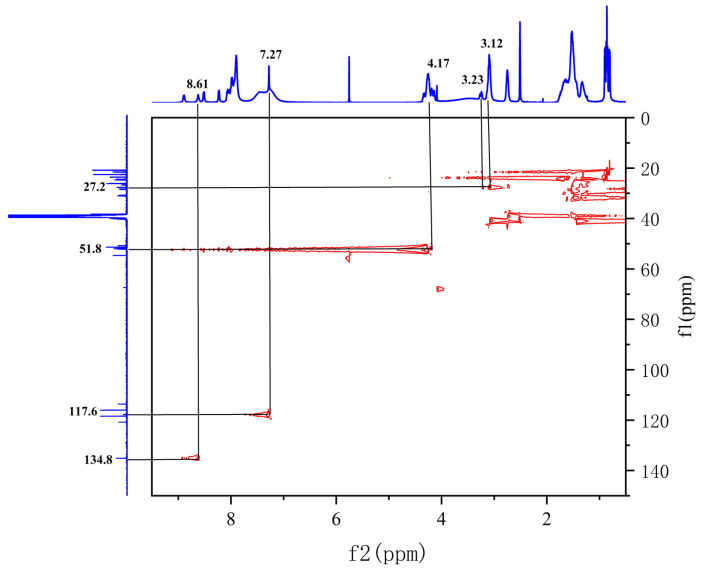
Spectrum of ^1^H-^13^C HSQC NMR spectrum of raw peptides.

**Figure 5 pharmaceuticals-17-00981-f005:**
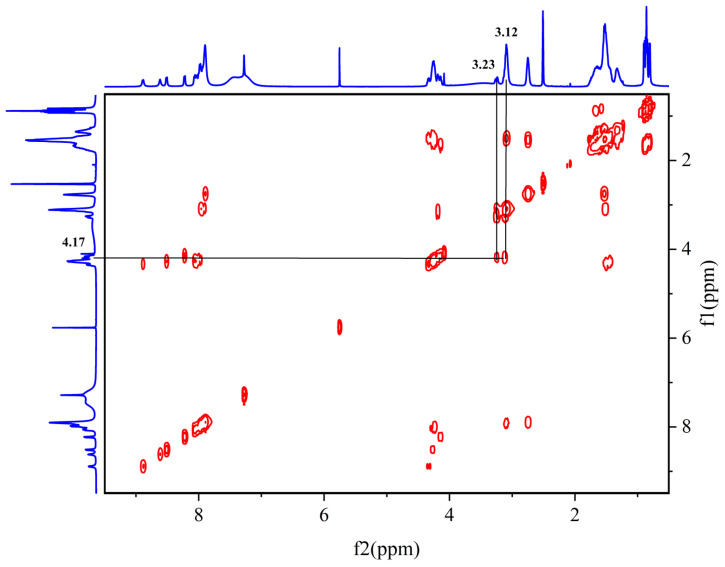
Spectrum of ^1^H-^1^H COSY NMR spectrum of raw peptides.

**Figure 6 pharmaceuticals-17-00981-f006:**
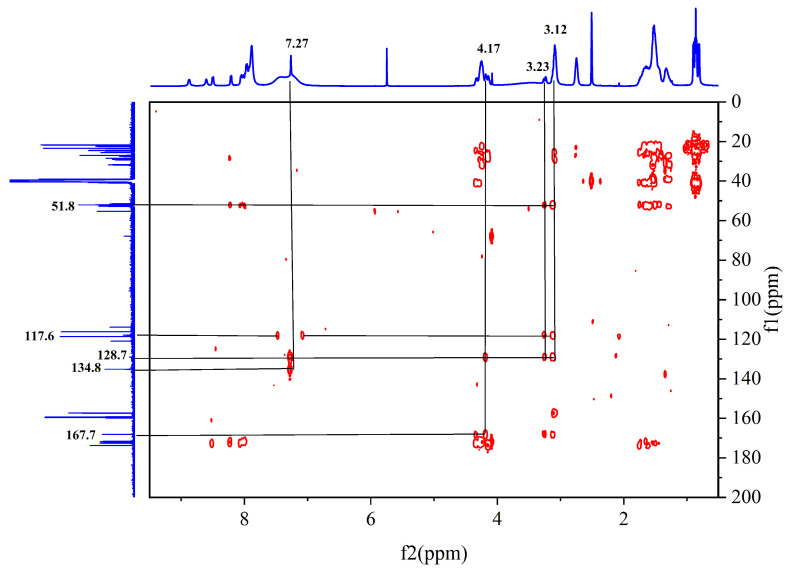
Spectrum of ^1^H-^13^C HMBC NMR spectrum of raw peptides.

**Figure 7 pharmaceuticals-17-00981-f007:**
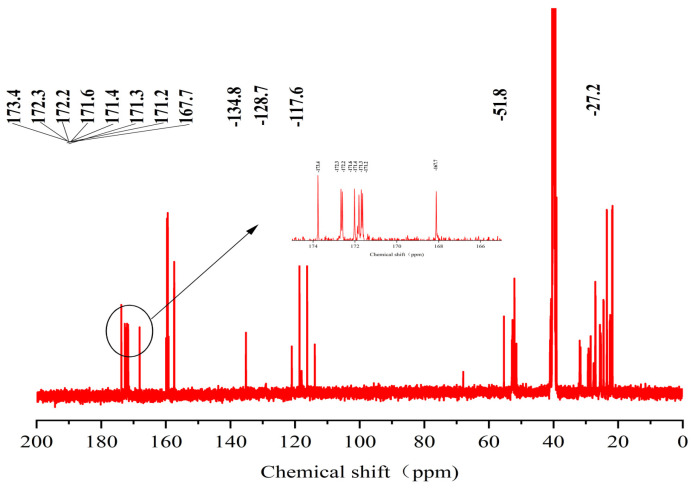
Spectrum of ^13^C-NMR spectrum of raw peptides.

**Figure 8 pharmaceuticals-17-00981-f008:**
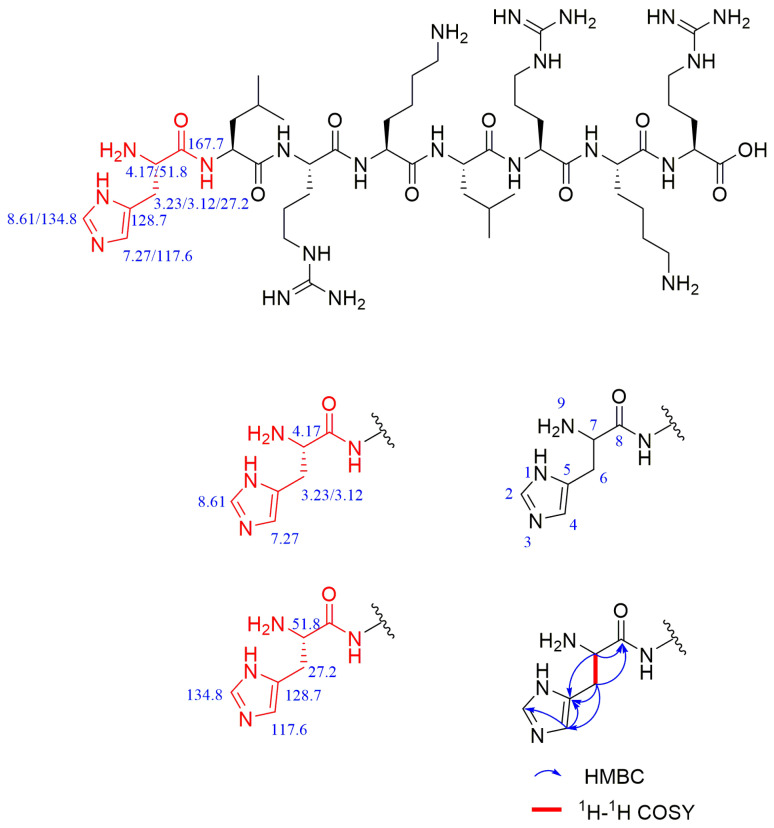
Chemical shifts of important structures in raw polypeptides.

**Figure 9 pharmaceuticals-17-00981-f009:**
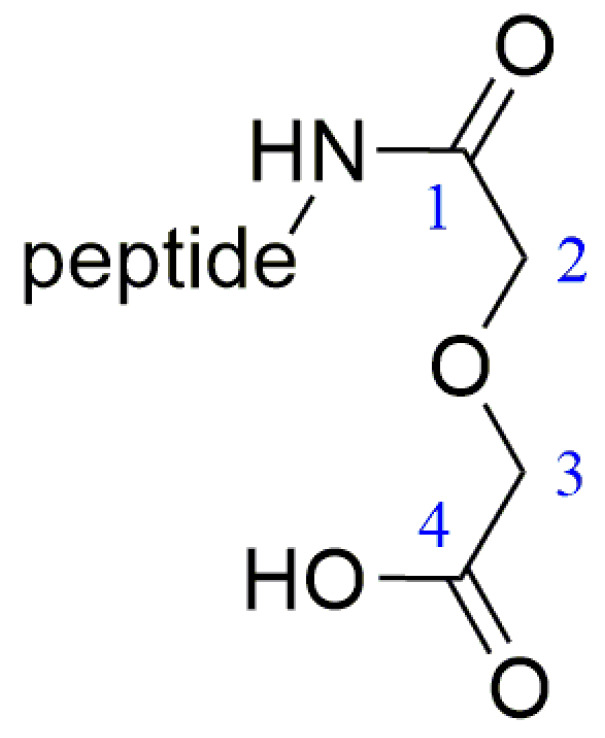
Coupling functional group label.

**Figure 10 pharmaceuticals-17-00981-f010:**
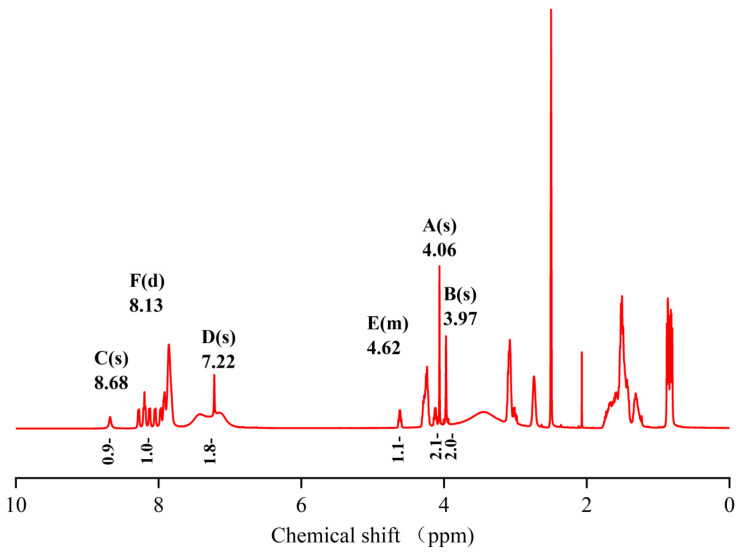
Spectrum of ^1^H-NMR spectrum of peptide product coupled to an anhydride molecule.

**Figure 11 pharmaceuticals-17-00981-f011:**
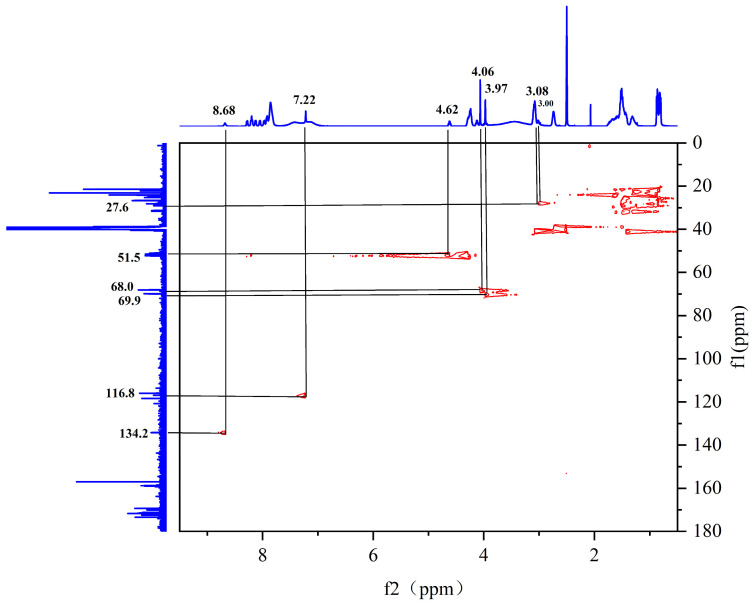
Spectrum of ^1^H-^13^C HSQC NMR spectrum of peptide product coupled to an anhydride molecule.

**Figure 12 pharmaceuticals-17-00981-f012:**
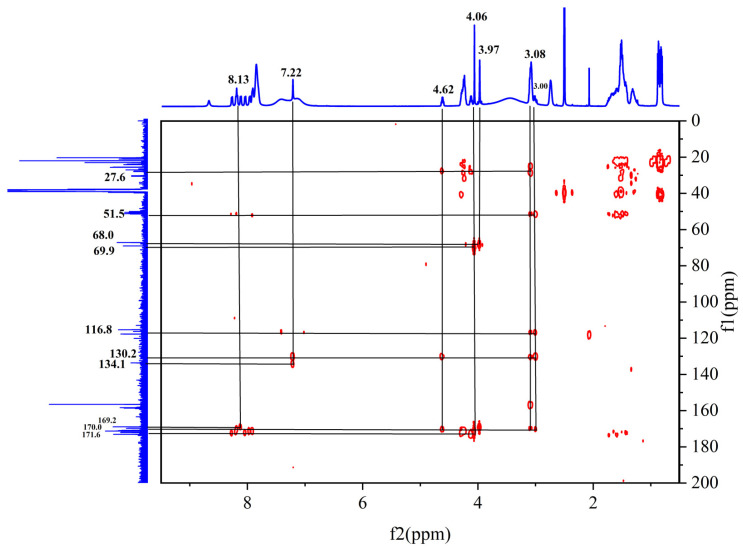
Spectrum of ^1^H-^13^C HMBC NMR spectrum of peptide product coupled to an anhydride molecule.

**Figure 13 pharmaceuticals-17-00981-f013:**
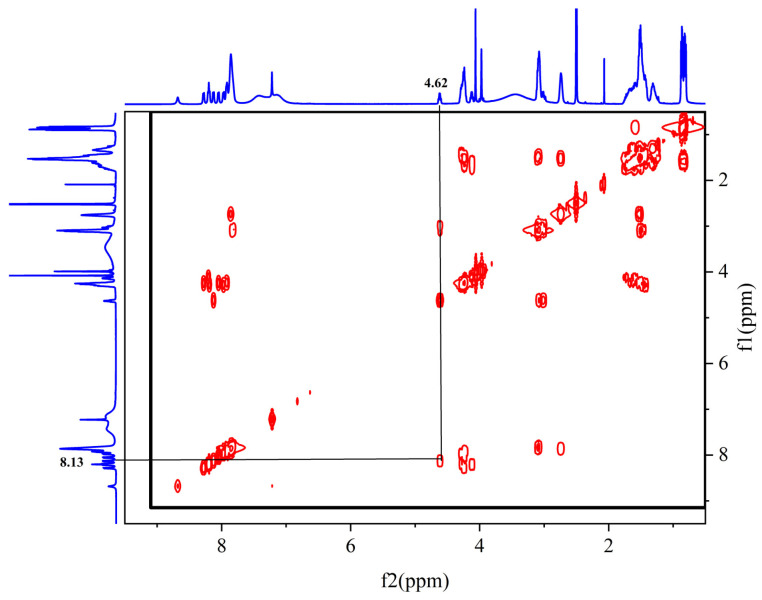
Spectrum of ^1^H-^1^HCOSY NMR spectrum of peptide product coupled to an anhydride molecule.

**Figure 14 pharmaceuticals-17-00981-f014:**
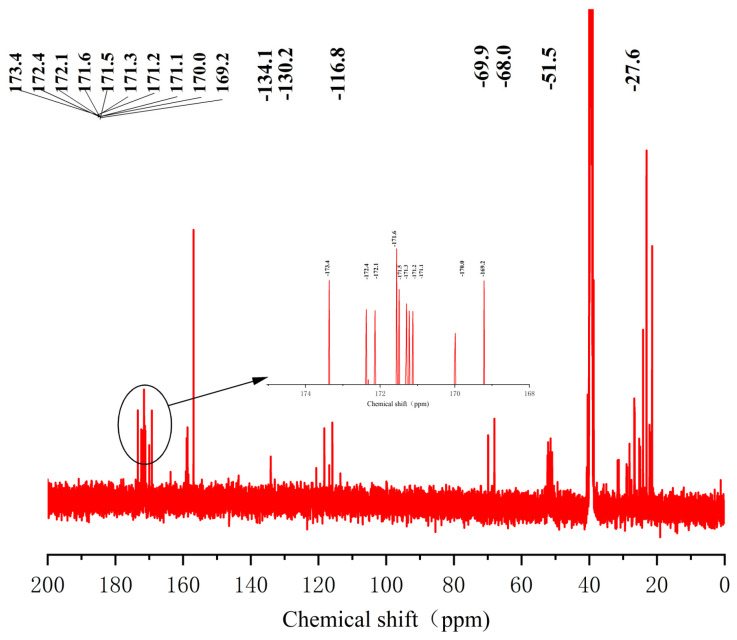
Spectrum of ^13^C-NMR spectrum of peptide product coupled to an anhydride molecule.

**Figure 15 pharmaceuticals-17-00981-f015:**
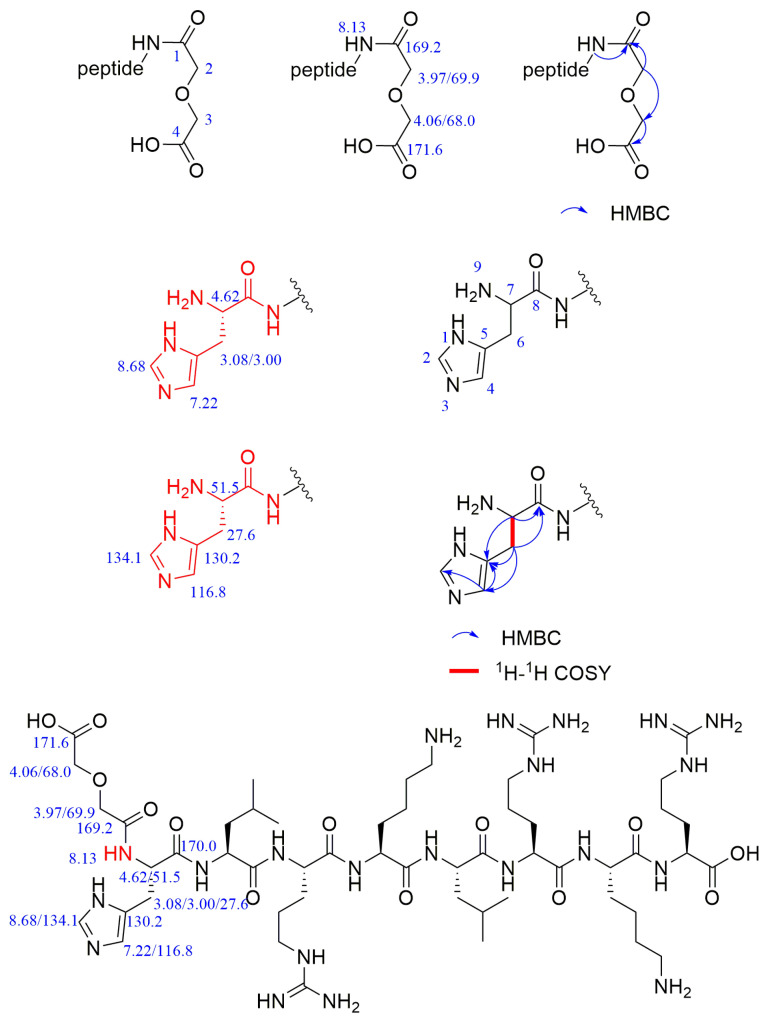
The chemical shift values of important structures in peptide products coupled with an anhydride molecule.

**Figure 16 pharmaceuticals-17-00981-f016:**
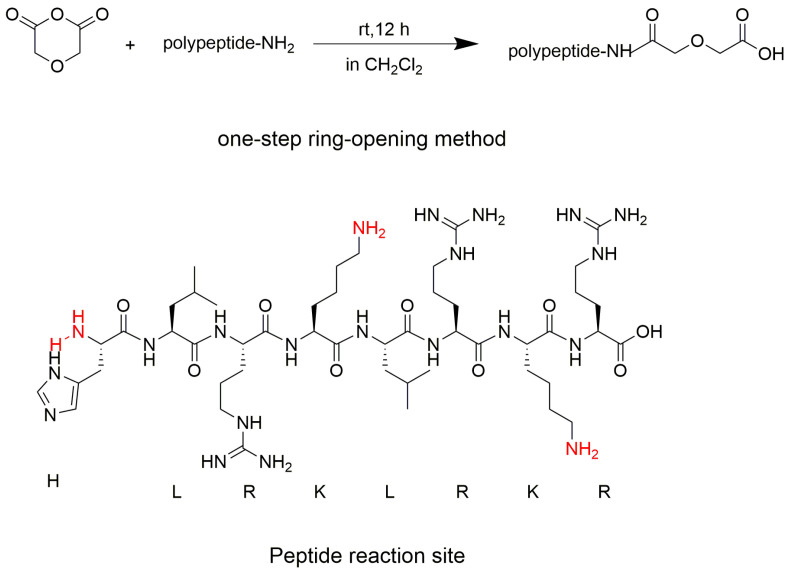
The structure of the polypeptide with associated equations.

**Table 1 pharmaceuticals-17-00981-t001:** Peptides:anhydride 1:10, individual peak areas.

Peak Departure Time (min)	Peak Area
10.226	3.71 × 10^4^
12.357	2.21 × 10^4^
14.43	1843.00
14.792	1980.03
16.791	301.90

**Table 2 pharmaceuticals-17-00981-t002:** Peptides:anhydride 1:20, individual peak areas.

Peak Departure Time (min)	Peak Area
11.034	2.12 × 10^4^
12.781	3.51 × 10^4^
14.667	4965.46
15.024	5633.18
17.027	1162.03

**Table 3 pharmaceuticals-17-00981-t003:** Peptides:anhydride 1:40, individual peak areas.

Peak Departure Time (min)	Peak Area
10.529	8967.82
10.461	3.68 × 10^4^
14.148	1.12 × 10^4^
14.503	1.34 × 10^4^
16.587	5453.32

**Table 4 pharmaceuticals-17-00981-t004:** Peptides:anhydride 1:60, individual peak areas.

Peak Departure Time (min)	Peak Area
10.597	8070.40
12.378	2.10 × 10^4^
14.201	1.20 × 10^4^
14.553	1.56 × 10^4^
15.771	1.58 × 10^4^

**Table 5 pharmaceuticals-17-00981-t005:** Conversion of peptides under different equivalent anhydride conditions.

Peptide to Anhydride Equivalent Ratio	Raw Material Conversion Rate
1:10	43%
1:20	67%
1:40	86%
1:60	87%

**Table 6 pharmaceuticals-17-00981-t006:** Peptide modification results.

Peak Departure Time (min)	Modifications	Modified Sequence	Mass	Intensity
HLRKLRKR	C_4_H_4_O_4_(HKR)	H(c4)LRKLRKR	1221.7418	4.07 × 10^11^
HLRKLRKR	2C_4_H_4_O_4_(HKR)	H(c4)LRKLRK(c4)R	1337.7528	6.18 × 10^11^
HLRKLRKR	2C_4_H_4_O_4_(HKR)	H(c4)LRK(c4)LRKR	1337.7531	3.57 × 10^11^
HLRKLRKR	3C_4_H_4_O_4_(HKR)	H(c4)LRK(c4)LRK(c4)R	1453.7637	7.78 × 10^10^

## Data Availability

The data present in this study are available in the body of the manuscript.

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
