# Peer review of "Development of a Novel, Easy-to-Prepare, and Potentially Valuable Peptide Coupling Technology Utilizing Amide Acid as a Linker"

_pharmaceuticals, 2024, doi:10.3390/ph17080981_

Round 1

Reviewer 1 Report

Comments and Suggestions for Authors

This manuscript introduces diglycolic anhydride as a coupling reagent and investigates its reaction position with a single peptide. The manuscript requires significant polishing, as there are duplicated paragraphs or descriptions (e.g., lines 94-97). Additionally, it lacks important information that needs to be included before submission.

1. In the Results and Discussion section, the authors used 2D NMR to verify that the anhydride attaches to the N-terminal amine group of the peptide. However, there is no evidence showing how the anhydride couples with the amine groups of the Lys side chain, except for the mass spectrometry data. It is unclear where the results presented in Table 6 originate.

2. The reaction between diglycolic anhydride and amines is not novel. Kit Lam and other researchers have utilized this reaction in peptide coupling and linker formation for a long time. As depicted in Figure 2, the reaction of diglycolic anhydride with the peptide does not yield a high product (approximately 50%) and is not very clean, producing several byproducts. 

3. In the Abstract and Introduction sections, the authors repeatedly emphasize that the product of diglycolic anhydride with peptide can be used as a chelating functional group and is useful for binding lanthanide ions. However, there are no related experiments in the subsequent research.

4. The authors should provide detailed NMR experimental methods in Section 2-Materials and Methods.

Reviewer 2 Report

Comments and Suggestions for Authors

The article title Development of a novel, easy-to-prepare, and potentially valuable peptide coupling technology utilizing amide acid as a linker is accepted after consideration of the following comments.

1)    Abstract, he results of new reactions should be mentioned regarding the time and yield.

2)    Abstract, we, i, us should not be used.

3)    Introduction, line 71, 72 NH2 should change to NH2

4)    Line 84, as11C should change to as 11C.

5)    Line 86. .Reaction to . Reaction

6)    Rational for this study should be improve and the economic or biological feedback of this study should be mentioned in details.

7)    Figure 1, What is CH2C12.

8)    Table 5, table 5 the differences between 1:40 and 1 : 60 is very small so the amount of anhydride can be reduced to 41 to 59 %to show the maximum amount of anhydride make raw material conversion to 87 .

9)    Conclusion should be brief, interpret the results and suggest future plan for this study.

10)                    References should update as only one 2023

Reviewer 3 Report

Comments and Suggestions for Authors

The article titled Development of a novel, easy-to-prepare, and potentially valuable peptide coupling technology utilizing amide acid as a linker by Yaling Wang, Fan Yang and Hongyan Li is concerned with the synthesis and spectroscopic analysis of designed a novel bipotent chelator coupled with peptides. The authors used a one-step ring-opening method to introduce the coupling functional group of the tridentate amide acid, and the coupling material included the amino active site of the peptide and diethylene glycol anhydride. In this paper, the researchers explored the reactions between different equivalents of acid anhydride coupled to the peptide (peptide sequence: HLRKLRKR) and determined that the maximum conversion of the peptide feedstock was 87%. To determine the selectivity of the reaction sites in this polypeptide, they use nuclear magnetic resonance (NMR) and liquid chromatography-mass spectrometry (LC-MS).

The abstract should be changed, because in the manuscript the authors do not obtain any radionuclide.  In 2.3 section, what is the solvent of CH2C12?

I am confused with the ring-opening process - there is a need to write more synthetic details.

What is the solvent for the NMR analysis?

The "structural structure" of the polypeptide with associated equations shown in Figure 1 should be changed because what does structural structure mean? Maybe molecular structure?

In Figure 4, what are the capitalized letters labelled on the spectra? This information is not explained in the text.

What is the f1 and f2? There should be written the type of spectrum on the figure and the correlation between them.

In Figure 7, the line should be proofread as the researchers have written “the H-6 region showed a stable correlation with C-7 (51.8 ppm)”

The sentence is not correct in connection to Figure 13.

“1H 1HCOSY spectra (Figure 13), a clear correlation signal between H-7(4.17 ppm) and H-6 (3.23/3.12 ppm) can be observed” it is not the truth in this case; in Figure 13 I can not see this information.  

Whole text needs editorial correction, the text should be checked carefully and proofread.

Round 2

Reviewer 3 Report

Comments and Suggestions for Authors

The submitted manuscript, after the corrections made, I believe is suitable for publication in the Pharmaceuticals journal.

The article is more readable.